# Impact of Advanced Age on the Clinical Presentation and Outcome of Sporadic Medullary Thyroid Carcinoma

**DOI:** 10.3390/cancers13010094

**Published:** 2020-12-30

**Authors:** Antonio Matrone, Carla Gambale, Alessandro Prete, Paolo Piaggi, Virginia Cappagli, Valeria Bottici, Cristina Romei, Raffaele Ciampi, Liborio Torregrossa, Luigi De Napoli, Eleonora Molinaro, Gabriele Materazzi, Fulvio Basolo, Rossella Elisei

**Affiliations:** 1 Endocrine Unit, Department of Clinical and Experimental Medicine, University Hospital of Pisa, Via Paradisa 2, 56124 Pisa, Italy; antonio.matrone@med.unipi.it (A.M.); carla.gambale@phd.unipi.it (C.G.); a.prete1@studenti.unipi.it (A.P.); virginia.cappagli@med.unipi.it (V.C.); valeria.bottici@ao-pisa.toscana.it (V.B.); cristina.romei@unipi.it (C.R.); raffaele.ciampi@unipi.it (R.C.); e.molinaro@ao-pisa.toscana.it (E.M.); 2National Institute of Diabetes and Digestive and Kidney Diseases, National Institutes of Health, Phoenix, AZ 85016, USA; paolo.piaggi@unipi.it; 3Department of Information Engineering, University of Pisa, 56124 Pisa, Italy; 4Department of Surgical, Medical, Molecular Pathology and Critical Area, University Hospital of Pisa, 56124 Pisa, Italy; l.torregrossa@ao-pisa.toscana.it (L.T.); fulvio.basolo@med.unipi.it (F.B.); 5Unit of Endocrine Surgery, Department of Surgical, Medical, Molecular Pathology and Critical Area, University Hospital of Pisa, 56124 Pisa, Italy; l.denapoli@hotmail.it (L.D.N.); gabriele.materazzi@med.unipi.it (G.M.)

**Keywords:** medullary thyroid carcinoma, sporadic, age, distant metastasis, death rate

## Abstract

**Simple Summary:**

The clinical behavior of medullary thyroid carcinoma is heterogeneous and can be influenced by several clinical, biochemical and molecular factors. The role of age as a prognostic factor remains controversial. In our cohort of 432 sporadic medullary thyroid carcinoma, no differences in histologic features at diagnosis and in number and type of therapies performed during the follow-up were detected when dividing the patients according to age (< and ≥ 65 years). Younger patients had a longer follow-up and survival time, compared to the older patients. However, in dead patients, no differences in the aggressiveness of the disease at presentation and treatments performed during the follow-up were found between the two age groups.

**Abstract:**

Sporadic medullary thyroid carcinoma (MTC) is a rare malignancy with a heterogeneous clinical course. Several potential prognostic factors have been investigated, but the impact of some of these is controversial, such as age at diagnosis. We evaluated the data of 432 sporadic MTC patients followed-up for a median of 7.4 years. Patients were divided and compared according to their age at diagnosis in group A (<65 years—n = 338, 78.2%) and group B (≥65 years—n = 94, 21.8%). No differences were detected between the two groups. Median follow-up time was significantly longer in patients <65 than ≥65 years. We observed 41 (9.5%) cancer-related death events. The death rate was similar between the two age groups. However, the Kaplan Meier curve showed a longer survival time for younger patients compared to older patients [HR 2.5 (CI 95%: 1.27–4.94), *p* < 0.01]. Nevertheless, no differences in the aggressiveness of the disease at presentation and in the number and type of treatments performed were found in the two subgroups of dead patients. In patients with sporadic MTC, age at diagnosis did not correlate with any clinical and pathological features. Cancer-related death events are similar in older and younger patients, but survival time is longer in the younger.

## 1. Introduction

Medullary thyroid carcinoma (MTC) is a rare malignancy [1] occurring in sporadic or hereditary form. The clinical course and outcome are heterogeneous, and several factors can influence the prognosis. Female sex [2] and earlier stages of the disease [2,3] showed better prognosis. Conversely, histologic features, such as extra-thyroidal extension, lymph node involvement and distant metastasis [4,5,6], could negatively influence the clinical course of the disease. Likewise, high calcitonin (CT) values [7] and aggressive REarranged during Transfection (*RET*) gene mutation (i.e., M918T) [8] have been demonstrated as negative prognostic factors.

Although the impact of age in the prognosis of MTC has been evaluated in several papers [3,9,10], the results remain controversial. In most of the cases, both sporadic and hereditary MTC were included, thus confounding the interpretation of the data.

The aim of our study was to evaluate the presentation and clinical outcome of sporadic MTC in the elderly (≥65 years) [11]. Then, we compared epidemiological, clinical and pathological data obtained with a younger population (<65 years).

## 2. Results

Epidemiologic, clinical and pathologic data of the study population are summarized in Table 1.

Females were more numerous (56.5%), and the median age at the diagnosis was 54 years. The pre-operative CT value was available in 358/432 (82.9%) patients and in most of them (58.9%) was higher than 100 pg/mL.

In 74 cases, the pre-operative CT value was not available, because of patients who were not surgically treated in Pisa and missed the pre-operative diagnostic work-up, or the diagnosis of MTC was incidental (discovered only after surgery).

More than half of the cases (55.3%) showed tumor diameters between 1.1 and 4 cm; in 36.1% a microcarcinoma was observed, while in the remaining cases (6.6%) tumor diameter was larger than 4 cm. The tumor was unifocal in 84% of cases, and mETE was present in about 20% of the cases.

Central and/or latero-cervical compartment lymph node dissection was performed in 87.5% and 36.6% of cases, respectively. Excluding 33 patients (7.6%) in which lymph nodes were not removed (Nx), most of the remaining had no lymph node metastases (N0 47.2%). Conversely, 66 patients (15.3%) showed central compartment (N1a), 36 (8.3%) latero-cervical compartment (N1b) and 93 (21.5%) had simultaneous central and latero-cervical compartment lymph node metastasis (N1a + N1b).

In 264/432 (61.1%) patients, histological variants of MTC were analyzed. Conventional MTC was the most frequent variant (59.5%), followed by spindle cell (21.6%). Moreover, the presence of somatic mutations was investigated in a subgroup of 160 (37%) patients. RET M918T was the most frequent mutation.

During the follow-up, 48 (11.1%) patients were submitted to a second neck surgery, 44 (10.2%) to local treatments and 59 (13.7%) to systemic treatments.

After a median follow-up of 7.4 years (88.5 months), 229 (53%) patients showed an excellent response, 67 (15.5%) a biochemical incomplete response and 136 (31.5%) had metastatic disease of whom 21.3% (29/136) at local and 78.7% (107/136) at distant sites.

### 2.1. Differences in Sporadic MTC Patients According to Age < or ≥ 65 Years

As for the aim of the study, patients were divided according to their ages at diagnosis in group A (<65 years—n = 338, 78.2%) and group B (≥65 years—n = 94, 21.8%) (Table 2).

As expected, the median age of the two groups was different (49 vs 70 years, *p* < 0.01). Conversely, no differences in sex distribution (females 57.1% vs 54.3%, *p =* 0.62) were shown.

Clinical presentation of the disease was the same in the two groups, in particular median tumor dimension (1.5 vs 1.6 cm, *p* = 0.70), the prevalence of microcarcinoma (36.4% vs 38.5%, *p* = 0.81), multifocality (15.7% vs 17%, *p* = 0.75) and the presence of mETE (19.8% vs 20.2%, *p* = 0.93) showed no differences between group A and B.

Histological variants of MTC showed the same prevalence in the two groups (*p* = 0.2).

Regarding TNM, neither T stage (*p* = 0.92) nor N stage (*p* = 0.91) and M stage (*p* = 0.13) differed between the groups.

Moreover, the prevalence of central and latero-cervical compartment lymph node dissection was the same between the two groups (87.6% vs 87.2%, *p* = 0.93 and 37.3% vs 34%, *p* = 0.56, respectively), setting to zero the potential interference of the surgical treatment performed on the N result.

No difference in the prevalence (*p* = 0.22) and type (*p* = 0.28) of *RET* somatic mutations was found between group A and B.

Also, additional surgical local and systemic treatments were performed in the same percentage of patients in the two groups.

Although group B had a median follow-up significantly shorter than group A (61 vs 96 months, *p* < 0.01), clinical outcome did not differ (*p* = 0.16).

### 2.2. Cancer-Related Death Events

In our study group, we observed 41 (9.5%) cancer-related death events. All of these patients were included in the distant metastatic group (41/136—30.1%) and death events were related to the progression of the disease. When dividing these patients according to age at surgery, 29/338 (8.6%) were in group A and 12/94 (12.8%) were in group B, and this difference was not statistically significant (*p* = 0.22).

However, the Kaplan-Meier analysis (Figure 1) showed a significant difference in the survival rate between the two groups [HR 2.5 (CI 95%: 1.27–4.94), *p* < 0.01], with the rate of survival being higher in patients of group A.

Therefore, we assessed the potential impact of the aggressiveness of the presentation of the disease in the two subgroups of patients who died (Table 3).

However, no significant differences were found, suggesting a minimal impact of the presentation of the disease in cancer-related death events when patients were divided according to age.

Moreover, when we evaluated the time elapsed between surgery and death (≤5 yrs, 5.1–10 yrs, 10.1–15 yrs and >15 yrs), again no statistically significant differences were shown (Table 4).

Furthermore, when dividing the age at diagnosis in quartiles (≤43 yrs, 44–54 yrs, 55–63 yrs and ≥64 yrs), the Kaplan-Meier survival curves continued to show a slightly significant difference among the groups (*p* = 0.049) (Figure 2). Again, patients older than 64 years showed worse survival over time.

## 3. Discussion

Medullary thyroid carcinoma is a rare malignancy representing 1–2% of thyroid cancers in the USA [1].

In about 15–20% of cases, it can be hereditary due to a germline *RET* mutation [12]. The remaining 80% of cases are sporadic, and more than half of them can harbor a somatic *RET* mutation [13]. The ability of MTC to metastasize both by blood and lymphatic vessels makes the prognosis of this tumor intermediate between the quite good prognosis of differentiated thyroid cancer (DTC) and the poor prognosis of anaplastic ones [3,14]. Several epidemiologic and clinic-pathologic factors can contribute to the heterogeneous clinical course of MTC [15,16]. Age is a determinant in the prognosis of DTC [17], indeed also in the AJCC (American Joint Committee on Cancer) staging system [18]; different ages characterize different stages with different prognostic values, even in patients with the same tumor burden. This is not the case for MTC [18]. While in hereditary MTC the role of age is crucial [19], nothing is known about its impact on the clinical presentation and prognosis in sporadic MTC.

In our study, we did not find any differences in the clinical presentation of MTC in the two groups of patients divided by age. Moreover, no differences were found when we looked at the number and types of further local or systemic treatments performed in the two groups.

Several studies analyzed the impact of age on the clinical features and outcome of MTC [4,5,10]. Sahli et al. [10] analyzed the SEER (Surveillance, Epidemiology, and End Results Program) database and evaluated histologic features and disease specific mortality in 1457 hereditary and sporadic MTC patients, according to 3 categories of age: younger adults (18–64 years), older adults (65–79 years) and super-elderly (≥80 years). Although they found significant differences in the extent of surgery (less aggressive in the elderly), in agreement with our results no differences were shown in the prevalence of lymph nodes and distant metastasis among the age groups.

Conversely, other authors found a correlation between older age and structural disease. Kotwal et al. [4] showed that an age of >55 years was significantly associated with the presence of distant metastasis at diagnosis. Hamdy et al. [5], in a cohort of 31 hereditary and sporadic MTC, showed that an age of >40 years increased 12-fold the risk of having distant metastasis, and was associated with shorter disease-free survival. Twito et al. [6] showed that age >45 years at diagnosis was associated with biochemical and/or structural persistent disease in a cohort of 193 MTC (of whom 18.1% were hereditary), at the time of last evaluation (median 7 years). However, when considering sporadic cases, only the presence of distant metastasis at diagnosis was associated with disease persistence.

It is worth noting that all previous studies included both hereditary and sporadic MTC. The different pathogenesis of these two forms should be considered when analyzing age as a prognostic factor, since the development of MTC in hereditary cases is much quicker than in sporadic ones, involving even children [19]. For this reason, we decided to perform this study only in sporadic cases to minimize the possibility of potential bias.

Cancer-related death in the whole cohort (9.5%) was similar to other published data [10] and the death rate did not differ between the two groups (8.6% in group A vs 12.8% in group B, *p* = 0.22).

However, the Kaplan-Meier analysis showed a longer survival time in younger patients who also had a significantly longer follow-up time, both when dividing the patients into two groups of age (<65 and ≥65 years) and when patients were divided into quartiles of age (≤43 yrs, 44–54 yrs, 55–63 yrs and ≥64 yrs). At the time of death, all patients, independently from age, had distant metastatic disease and died from disease progression, regardless of local or systemic treatments performed.

Age was analyzed as a potential prognostic factor of death in several papers including hereditary and sporadic MTC. In a multivariate analysis, both age and stage were predictive factors of survival in a cohort of 899 hereditary and sporadic MTC, included in the French Calcitonin Tumors Study Group (GETC) [3]. The same results were found by Kebebew et al. [9] in 104 patients with MTC or C-cell hyperplasia. Also, more recent papers confirmed this association. Kotwal et al. [4] showed that age >55 years was a significant predictor of worse overall and disease specific survival. In the paper of Sahli et al. [10], older adults and the super-elderly showed a disease specific mortality 2.9 and 6.7 times higher than younger adults, respectively.

At variance with other studies, in two subgroups of our patients, we even analyzed the prevalence of *RET* somatic mutation and the different histological variants of MTC. It is known that the presence of a *RET* somatic mutation is a poor prognostic factor for both the initial aggressiveness and survival of sporadic MTC [20]. Similarly, more aggressive behavior could be related to different histological variants [21]. However, neither *RET* mutations nor histological variants prevalence differed between the groups A and B.

We did not find any difference in the aggressiveness of presentation of the disease in the subgroup of MTC patients who died from the disease. Furthermore, by breaking down the follow-up time into intervals of 5 years, no difference in death rate between the two groups was shown. It is well known that older patients with cancers have a worse outcome per se related to several factors. Cancer more frequently occurs in older age [22], and in several cancers older patients receive less aggressive treatments [23,24]. Moreover, comorbidities increase with age, and, in subjects with the same age, patients with cancer show more comorbidities compared to those without [25,26]. Moreover, cancer patients with comorbidities usually receive less aggressive treatments [27].

In this scenario, although we observed that tumor burden at diagnosis did not differ between the two groups, we can hypothesize that older patients could have a pre-clinical disease for a longer time compared with younger patients, with an uncertain impact of this pre-clinical period of time on the final outcome.

Moreover, since no differences were shown in disease aggressiveness at presentation and number and types of treatments performed in dead patients, we can also hypothesize that the different survival times could be not necessarily related to the disease but to older age per se.

## 4. Materials and Methods

We retrospectively evaluated the epidemiologic, clinical and pathologic data of 530 consecutive sporadic MTC patients, surgically treated between 2000 and 2018 and followed-up at the Endocrine Unit of the Pisa University Hospital. For the purpose of this study, we excluded patients who were followed for less than 2 years (n = 98), resizing the study population to 432 cases.

The study was approved by the local Ethical Committee (CEAVNO—Comitato Etico Area Vasta Nord-Ovest; protocol number 57877), and, for the policy of the University Hospital, all patients provided a written informed consent to use their data for scientific purposes.

All patients were treated with total thyroidectomy. Central compartment lymph node dissection was routinely performed in most of the MTC patients except those incidentally discovered after surgery. Latero-cervical compartment lymph node dissection was performed only in patients with pre- or intra-operative evidence of latero-cervical lymph node metastasis. Most of the patients (323/432—74.8%) were surgically treated at the Endocrine Surgery Unit of the Pisa University Hospital. In the others, a revision of the histologic specimens was made by our specialized endocrine pathologists.

For each patient, we assessed sex and age at the diagnosis, the value of pre-operative basal serum calcitonin, TNM stage (according to AJCC 8th edition) and other histologic features such as tumor size, presence of minimal extrathyroidal extension (mETE) and multifocality. Moreover, further treatments after primary surgery (second neck surgery, local treatments, systemic treatments) and clinical outcome at the time of the last evaluation were reported.

We defined the clinical outcome as excellent response, when patients showed basal and/or stimulated CT values lower than the upper limit of the normal range, without evidence of structural disease; and biochemical incomplete response when basal and/or stimulated CT values were higher than the upper limit of normal range, without evidence of structural disease. Moreover, we divided the structural incomplete response patients into those who had local disease (cervical lymph node metastasis or local recurrences) and those who had distant metastatic disease, regardless of CT values.

### 4.1. CT Assays

All patients performed periodic clinical evaluation according to good clinical practice. At each evaluation, patients performed serum CT measurements and neck US as well as thyroid hormones to assess the adequacy of L-T4 substitutive therapy. During the study period (2000–2018), we changed two CT assays. From 2000 to 2013, serum CT was measured using an immunoradiometric assay (ELSA-hCT, CIS, Gif-Sur-Yvette, France) with a functional sensitivity of 10 pg/mL. From 2014 to now, CT was measured using chemiluminescent immunometric assay (Immulite, Siemens Healthcare Diagnostic Products Ltd., Lianberis, Gwynedd LL55 4EL, UK), with an analytical sensitivity of 2 pg/mL and an upper limit of normal range of 11.5 pg/mL in females and 18.2 pg/mL in males.

To define clinical response after surgery, in several cases, in addition to the basal CT, a stimulation test for CT was performed. Until 2013, a CT stimulation test was performed with the injection of pentagastrin (Pg) (Peptavlon, Nova Laboratories, LTD, Leicester UK, 0.5 mg/kg i.v.) and blood was collected before and 2, 5, 15, and 30 min after Pg injection. From 2014, the test was performed with the infusion of calcium (2.3 mg/Kg of calcium element), and blood was collected before and 2, 5 and 10 min after calcium infusion.

### 4.2. Histological Analysis

MTC can exhibit different histopathological architectures [21]. A tumor showing the typical solid or trabecular growth pattern has been defined as conventional. Alternatively, other less frequent architectural variants have been noted, as the following: (i) spindle cell variant, when the tumor is almost exclusively composed by neoplastic cells with a spindle morphology arranged in interweaving fascicles; (ii) papillary variant, in which true papillae with fibrovascular stalks are lined by neoplastic cells; (iii) follicular variant, in which the neoplastic cells form follicles, resembling those of follicular tumors, containing eosinophilic colloid-like material; (iv) oncocytic variant, showing a prevalence of neoplastic cells with abundant granular eosinophilic cytoplasm; (v) clear cell variant, when the tumor is composed of cells with optically clear cytoplasm; (vi) angiosarcoma-like pattern, in which the neoplastic cells appear along vascular spaces resembling those observed in angiosarcomas.

### 4.3. Molecular Analysis

The sporadic nature of MTC was defined according to the absence of germline *RET* mutation, negative familial history of the disease and lack of laboratory test and imaging suggestive of other endocrine neoplasia.

Germline *RET* mutation investigations were performed in all patients after surgery on blood samples collected in EDTA (2–5 mL) and somatic *RET* mutations were searched for on fresh tumoral tissue collected at surgery and on paraffin embedded archival slides. DNA extraction was performed using the QIAamp DNA Mini Kit (QIAGEN, Hilden, Germany) or by an automated method (Maxwell 16 Instrument, Promega, Madison, Genes 2019, 10, 698, 3 of 12 WI, USA). *RET* mutation research was performed in eight exons (5, 8, 10, 11, 13, 14, 15 and 16) by direct sequencing following a previously reported protocol [28] and with an Ion S5 targeted sequencing NGS method using a custom panel as previously described [29].

### 4.4. Neck US

Neck US was performed with two devices over time, with a multifrequency 7.5 to 12 MHz linear transducer (AU 590 Asynchronous, and MyLab 50; Esaote Biomedica, Firenze, Italy). All neck lymph node compartments were evaluated. Suspected lesions were assessed by US-guided fine needle aspiration cytology with the evaluation of CT in the washing fluid of the needle. In addition to neck US, other imaging procedures (CT scan, MRI) were performed during the follow-up, if necessary.

## 5. Statistical Analysis

Categorical variables are expressed as counts and percentages, and continuous variables as median (IQR). Chi-square tests were used to evaluate the association between categorical variables. The Mann-Whitney U test was used for continuous numerical variables. The effect of age on survival was assessed by survival analysis using Cox’s proportional hazards model. Because the proportional hazards assumption was not met across the entire range of follow-up time, the survival analysis was limited to a follow-up time of <15 years when patients were divided into two groups according to age at diagnosis. A p-value of less than 0.05 was considered statistically significant. Analysis was performed with SPSS (version 20.0, IBM Corp., Armonk, NY, USA).

## 6. Conclusions

In this study, we demonstrated that MTC older patients (>65 years) showed a clinical presentation of the disease similar to the younger patients (<65 years). As far as cancer-related death events are concerned, the number of patients was similar in the two groups, but the time of survival was shorter in older patients as demonstrated by the Kaplan-Meyer analysis.

## Figures and Tables

**Figure 1 cancers-13-00094-f001:**
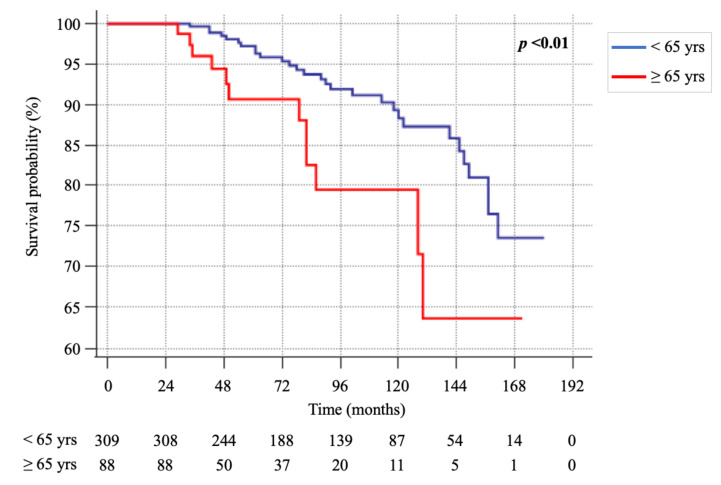
Kaplan-Meier survival curves of group A (<65 yrs) and B (≥65 yrs).

**Figure 2 cancers-13-00094-f002:**
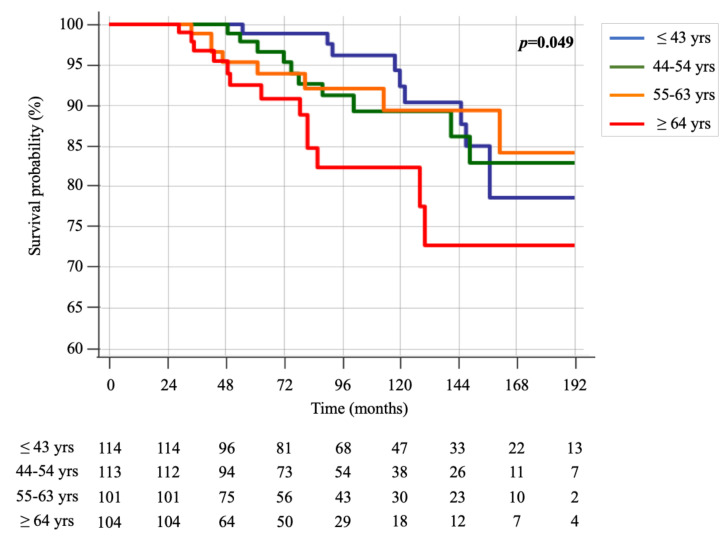
Kaplan-Meier survival curves of groups, divided into quartiles of age (≤43 yrs, 44–54 yrs, 55–63 yrs and ≥64 yrs).

**Table 1 cancers-13-00094-t001:** Epidemiologic, clinical and pathologic features of the 432 sporadic medullary thyroid carcinoma (MTC).

Features		n (%)
Sex	M	188 (43.5)
	F	244 (56.5)
Age (years)	Median (IQR)	54 (43–63)
	Min–Max	16–83
Pre-operative calcitonin ^a^	≤100 pg/mL	147 (41.1)
	>100 pg/mL	211 (58.9)
Tumor size ^b^ (cm)	Median (IQR)	1.5 (0.8–2.7)
	Min–Max	0.1–9
Tumor size category	≤1 cm	156 (36.1)
	1.1–4 cm	239 (55.3)
	>4 cm	28 (6.5)
Multifocality	Yes	69 (16)
	No	363 (84)
mETE	Yes	86 (19.9)
	No	346 (80.1)
Histologic pattern ^c^	Conventional	157 (59.5)
	Spindle Cell	57 (21.6)
	Others ^d^	50 (18.9)
T stage	T1a	147 (34)
	T1b	94 (21.8)
	T2	89 (20.6)
	T3	77 (17.8)
	T4	21 (4.9)
	Tx	4 (0.9)
Central compartment lymph node dissection	Yes	378 (87.5)
	No	54 (12.5)
Latero-cervical lymph node dissection	Yes	158 (36.6)
	No	274 (63.4)
N stage	N0	204 (47.2)
	N1a	66 (15.3)
	N1b	36 (8.3)
	N1a + N1b	93 (21.5)
	Nx	33 (7.6)
M stage	M0	22 (5.1)
	M1	48 (11.1)
	Mx	362 (83.8)
Somatic mutation investigation ^e^	Positive	94 (58.8)
	Negative	66 (41.3)
Somatic mutation result	RET M918T	63 (67)
	Others	31 (33)
Follow-up time (months)	Median (IQR)	88.5 (50–138.8)
	Min–Max	25–247
Second neck surgery	Yes	48 (11.1)
	No	384 (88.9)
Local treatments ^f^	Yes	44 (10.2)
	No	388 (89.8)
Systemic treatments	Yes	59 (13.7)
	No	373 (86.3)
Type of systemic treatment	Chemotherapy	5 (8.5)
	TKI	47 (79.7)
	Chemotherapy + TKI	5 (8.5)
	Somatostatin analogs	2 (3.4)
Clinical outcome	Excellent Response	229 (53)
	Biochemical Incomplete Response	67 (15.5)
	Local Metastatic disease	29 (6.7)
	Distant Metastatic disease	107 (24.8)
Site of surgery	Pisa	323 (74.8)
	No Pisa	109 (25.2)

^a^ available in 358/432—82.9%, ^b^ available in 423/432—97.9%, ^c^ available in 264/432—61.1%, ^d^ angiosarcoma, clear cell, follicular, mixed (papillary thyroid cancer/MTC), oncocytic, papillary, small cell, ^e^ available in 160/432—37%, ^f^ external beam radiotherapy (ERBT), trans-arterial chemioembolization (TACE), trans-arterial radioembolization (TARE).

**Table 2 cancers-13-00094-t002:** Comparison of analyzed features in group A (<65 yrs) and B (≥65 yrs).

Features		Group A (<65) (338—78.2%)	Group B (≥65) (94—21.8%)	*p*
Sex	M	145 (42.9)	43 (45.7)	0.62
	F	193 (57.1)	51 (54.3)	
Age (years)	Median (IQR)	49 (40–57)	70 (67–76)	<0.01
	Min–Max	(16–64)	(65–83)	
Pre-operative calcitonin ^a^	≤100 pg/mL	112 (40.4)	35 (43.2)	0.66
	>100 pg/mL	165 (59.6)	46 (56.8)	
Tumor size ^b^ (cm)	Median (IQR)	1.5 (0.8–2.5)	1.6 (0.8–2.8)	0.7
	Min–Max	0.1–9	0.2–8	
Tumor size category	≤1 cm	121 (36.4)	35 (38.5)	0.81
	1.1–4 cm	190 (57.2)	49 (53.8)	
	>4 cm	21 (6.3)	7 (7.7)	
Multifocality	Yes	53 (15.7)	16 (17)	0.75
	No	285 (84.3	78 (83)	
mETE	Yes	67 (19.8)	19 (20.2)	0.93
	No	271 (80.2)	75 (79.8)	
Histologic pattern ^c^	Conventional	117 (57.4)	40 (66.7)	0.2
	Spindle Cell	49 (24)	8 (13.3)	
	Others ^d^	38 (18.6)	12 (20)	
T stage	T1a	115 (34)	32 (34)	0.92
	T1b	77 (22.8)	17 (18.1)	
	T2	67 (19.8)	22 (23.4)	
	T3	59 (17.5)	18 (19.1)	
	T4	17 (5)	4 (4.3)	
	Tx	3 (0.9)	1 (1.1)	
Central compartment lymph node dissection	Yes	296 (87.6)	82 (87.2)	0.93
	No	42 (12.4)	12 (12.8)	
Latero-cervical lymph node dissection	Yes	126 (37.3)	32 (34)	0.57
	No	212 (62.7)	62 (66)	
N stage	N0	157 (46.4)	47 (50)	0.91
	N1a	54 (16)	12 (12.8)	
	N1b	28 (8.3)	8 (8.5)	
	N1a + N1b	72 (21.3)	21 (22.3)	
	Nx	27 (8)	6 (6.4)	
M stage	M0	21 (6.2)	1 (1.1)	0.13
	M1	38 (11.2)	10 (10.6)	
	Mx	279 (82.5)	83 (88.3)	
Somatic mutation investigation ^e^	Positive	81 (60.9)	13 (48.1)	0.22
	Negative	52 (39.1)	14 (51.9)	
Somatic mutation result	RET M918T	56 (69.1)	7 (53.8)	0.28
	Others	25 (30.9)	6 (46.2)	
Follow-up time (months)	Median (IQR)	96 (56–144.3)	61 (38–106)	<0.01
	Min–Max	25–247	25–242	
Second neck surgery	Yes	40 (11.8)	8 (85)	0.36
	No	298 (88.2)	86 (91.5)	
Local treatments ^f^	Yes	38 (11.2)	6 (6.4)	0.17
	No	300 (88.8)	88 (93.6)	
Systemic treatments	Yes	47 (13.9)	12 (12.8)	0.78
	No	291 (86.1)	82 (87.2)	
Type of systemic treatment	Chemotherapy	3 (6.4)	2 (16.7)	0.46
	TKI	39 (83)	8 (66.7)	
	Chemotherapy + TKI	4 (8.5)	1 (8.3)	
	Somatostatin analogs	1 (2.1)	1 (8.3)	
Clinical outcome	Excellent Response	175 (51.8)	54 (57.4)	0.51
	Biochemical Incomplete Response	57 (16.9)	10 (10.6)	
	Local Metastatic disease	23 (6.8)	6 (6.4)	
	Distant Metastatic disease	83 (24.6)	24 (25.5)	
Site of surgery	Pisa	250 (74)	73 (77.7)	0.47
	No Pisa	88 (26)	21 (22.3)	

^a^ available in 358/432—82.9%, ^b^ available in 423/432—97.9%, ^c^ available in 264/432—61.1%, ^d^ angiosarcoma, clear cell, follicular, oncocytic, papillary, small cell, ^e^ available in 160/432—37%, ^f^ external beam radiotherapy (ERBT), trans-arterial chemioembolization (TACE), trans-arterial radioembolization (TARE).

**Table 3 cancers-13-00094-t003:** Comparison of analyzed features in dead patients of group A and B.

Features		Group A Dead 29/338 (8.6%)	Group B Dead 12/94 (12.8%)	*p*
Sex	Male	21 (72.4)	10 (83.3)	0.46
	Female	8 (27.6)	2 (16.7)	
Age (years)	Median (IQR)	47 (41.5–58.5)	70.5 (67.3–74.3)	<0.01
	Min–Max	(26–64)	(65–79)	
Pre-operative calcitonin ^a^	≤100 pg/mL	2 (10)	-	0.28
	>100 pg/mL	18 (90)	11 (100)	
Tumor size ^b^ (cm)	Median (IQR)	2 (1.2–3.8)	3 (2.2–4.5)	0.17
	Min–Max	0.2–9	0.3–8	
Tumor size category	≤1 cm	5 (17.9)	1 (9.1)	0.69
	1.1–4 cm	18 (64.3)	7 (63.6)	
	>4 cm	5 (17.9)	3 (27.3)	
Multifocality	Yes	8 (27.6)	5 (41.7)	0.38
	No	21 (72.4)	7 (58.3)	
mETE	Yes	15 (51.7)	6 (50)	0.92
	No	14 (48.3)	6 (50)	
Histologic pattern ^c^	Conventional	3 (37.5)	3 (75)	0.07
	Spindle Cell	5 (62.5)	-	
	Others ^d^	-	1 (25)	
T stage	T1a	4 (13.8)	2 (16.7)	0.36
	T1b	4 (13.8)	-	
	T2	4 (13.8)	3 (25)	
	T3	14 (48.3)	4 (33.3)	
	T4	3 (10.3)	2 (16.7)	
	Tx	-	1 (8.3)	
Central compartment lymph node dissection	Yes	23 (79.3)	10 (83.3)	0.68
	No	6 (20.7)	2 (16.7)	
Latero-cervical lymph node dissection	Yes	24 (82.8)	11(91.7)	0.46
	No	5 (17.2)	1 (8.3)	
N stage	N0	3 (10.3)	1 (8.3)	0.92
	N1a	1 (3.4)	1 (8.3)	
	N1b	11 (37.9)	5 (41.7)	
	N1a + N1b	13 (44.8)	5 (41.7)	
	Nx	1 (3.4)	-	
M stage	M0	3 (10.3)	1 (8.3)	0.97
	M1	15 (51.7)	6 (50)	
	Mx	11 (37.9)	5 (41.7)	
Somatic mutation investigation ^e^	Positive	21 (87.5)	5 (71.4)	0.31
	Negative	3 (12.5)	2 (28.6)	
Somatic mutation result	RET M918	16 (76.2)	4 (80)	0.86
	Others	5 (23.8)	1 (20)	
Follow-up time (months)	Median (IQR)	84 (53.5–131.5)	61.5 (36–81.8)	0.08
	Min–Max	31–167	28–160	
Second neck surgery	Yes	12 (41.4)	4 (33.3)	0.63
	No	17 (58.6)	8 (66.7)	
Local treatments ^f^	Yes	14 (48.3)	6 (50)	0.92
	No	15 (51.7)	6 (50)	
Systemic treatments	Yes	21 (72.4)	9 (75)	0.87
	No	8 (27.6)	3 (25)	
Type of systemic treatment	Chemotherapy	2 (9.5)	1 (11.1)	0.91
	TKI	15 (71.4)	7 (77.8)	
	Chemotherapy + TKI	3 (14.3)	1 (11.1)	
	Somatostatin analogs	1 (4.8)	-	
Site of surgery	Pisa	11 (37.9)	7 (58.3)	0.23
	No Pisa	18 (62.1)	5 (41.7)	

^a^ available in 31/41—75.6%, ^b^ available in 39/41—95.1%, ^c^ available in 12/41—29.3%, ^d^ angiosarcoma, clear cell, follicular, mixed (papillary thyroid cancer/MTC), oncocytic, papillary, small cell, ^e^ available in 31/41—75.6%, ^f^ external beam radiotherapy (ERBT), trans-arterial chemioembolization (TACE), trans-arterial radioembolization (TARE).

**Table 4 cancers-13-00094-t004:** Death rate in patients of group A and B, divided by 5 years interval of follow-up.

Follow Up Time Interval (Years)		Group A n° (%)	Group B n° (%)	*p*
≤5	Alive	88 (89.8)	41 (87.2)	0.65
	Dead	10 (10.2)	6 (12.8)	
5.1–10	Alive	112 (90.3)	25 (86.2)	0.51
	Dead	12 (9.7)	4 (13.8)	
10.1–15	Alive	78 (91.8)	9 (81.8)	0.29
	Dead	7 (8.2)	2 (18.2)	
>15	Alive	31 (100)	7 (100)	-
	Dead	-	-	

## Data Availability

The data presented in this study are available on request from the corresponding author. The data are not publicly available due to patient privacy and the General Data Protection Regulation.

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
