# Peer review of "Impact of Advanced Age on the Clinical Presentation and Outcome of Sporadic Medullary Thyroid Carcinoma"

_cancers, 2020, doi:10.3390/cancers13010094_

Round 1
Reviewer 1 Report
The paper is well written and the results are clearly presented.
I have only a few comments.
The choice of 65 years is arbitrary. I'm not contrary to this choice but I suggest to 1) perform a ROC cirve analysis to find the best cut-off, and 2) to evaluate the patients by dividing them in 4 quartiles to search differences between the 4 groups.
In the figure 1 please replace "group A" with "<65 yrs" and "group B" with "≥ 65 yrs" to help the reader.
Reviewer 2 Report
very concise and sound paper
there are just a very few open issues:
Discussion section: medullary microcarcinoma was highly prevalent in the study population which might be most likely due to the intensive screening strategy commonly employed in Italy and, in particulary in the PISA University Hospital. Might CT screening have erased some adverse impact on outcome in elderly found in other studies?
Page 10 the authors describe extensively the methodology of pentagastrin testing, while providing no results. These results might be of interest with regard to the group of patients with incomplete biochemical remission. Are there any differences in Group A and B?
Figure 1: legend to x-axis should better read Time (months) instead of years. Please insert also the p-value denoting the finding of a significant statistical difference with regard to survial in Group A and B.
